# Behavior of Adult *Aedes aegypti* and *Aedes albopictus* in Kinshasa, DRC, and the Implications for Control

**DOI:** 10.3390/tropicalmed8040207

**Published:** 2023-03-30

**Authors:** Emile Zola Manzambi, Guillaume Binene Mbuka, Gillon Ilombe, Richard Mundeke Takasongo, Francis Wat’senga Tezzo, Maria del Carmen Marquetti, Emery Metelo, Veerle Vanlerberghe, Wim Van Bortel

**Affiliations:** 1Unit of Entomology, Department of Parasitology, National Institute of Biomedical Research, Kinshasa, Democratic Republic of the Congo; 2Global Health Institute, Faculty of Medicine, University of Antwerp, 2000 Antwerp, Belgium; 3Vector Control Department, Institute of Tropical Medicine ‘Pedro Kouri’, Havana, Cuba; 4Tropical Infectious Disease Group, Public Health Department, Institute of Tropical Medicine, 2000 Antwerp, Belgium; 5Outbreak Research Team, Institute of Tropical Medicine, 2000 Antwerp, Belgium; 6Unit of Entomology, Biomedical Science Department, Institute of Tropical Medicine, 2000 Antwerp, Belgium

**Keywords:** *Aedes*, exophagic, vector control, Democratic Republic of the Congo, Kinshasa

## Abstract

Yellow fever and chikungunya outbreaks—and a few dengue cases—have been reported in the Democratic Republic of the Congo (DRC) in recent years. However, little is known about the ecology and behavior of the adult disease vector species, *Aedes aegypti* and *Aedes albopictus*, in DRC. Preliminary studies showed important differences in *Aedes* behavior in DRC and Latin-American sites. Therefore, this study aimed to assess the host-seeking and resting behaviors of female *Ae. aegypti* and *Ae. albopictus*, and their densities in four communes of Kinshasa (Kalamu, Lingwala, Mont Ngafula and Ndjili). Two cross-sectional surveys were carried out, one in the dry season (July 2019) and one in the rainy season (February 2020). We used three different adult vector collection methods: BG-Sentinel 2, BG-GAT, and prokopack. Both *Aedes* species were clearly exophagic, exophilic, and sought breeding sites outdoors. The adult house index for *Ae. aegypti* exceeded 55% in all communes except Lingwala, where it was only 27%. The Adult Breteau Index (ABI) for *Ae. aegypti* was 190.77 mosquitoes per 100 houses inspected in the rainy season and 6.03 in the dry season. For *Ae. albopictus*, the ABI was 11.79 and 3.52 in the rainy and dry seasons, respectively. *Aedes aegypti* showed unimodal host-seeking activity between 6 h and 21 h. The exophagic and exophilic behaviors of both species point to the need to target adult mosquitoes outdoors when implementing vector control.

## 1. Introduction

Arboviruses such as dengue, yellow fever, chikungunya, and Zika are among the most important causes of emerging infectious diseases worldwide, but often go unnoticed in Africa [1,2]. Yellow fever and chikungunya outbreaks—and a few dengue cases—have been reported in recent years in the Democratic Republic of the Congo (DRC) [3,4,5,6,7,8]. Yellow fever outbreaks have occurred recently in Muanda (2016, Kongo Central Province) and Kenge (2019, Kwango Province), and a chikungunya outbreak occurred in Matadi (2019, Kongo Central Province) [9]. Although *Aedes aegypti* is the primary vector for the chikungunya virus, the outbreak in Matadi was driven by *Aedes albopictus*, an invasive species that might change the epidemiology of this and other arboviruses in DRC [9,10,11]. Although the main vector species, *Ae. aegypti* and *Ae. albopictus*, were reported in the DRC and incriminated in several arbovirus disease outbreaks, the lack of detailed information on their presence, ecology, behavior, and spread hampers understanding of transmission dynamics of the arboviruses in the country. In 2018, an entomological surveillance was conducted in Kinshasa to assess the infestation of immature *Aedes* stages (larvae and pupae). In all surveyed communes (Kalamu, Lingwala, Mont Ngafula and Ndjili), the larval indices were above the World Health Organization arbovirus transmission thresholds, indicating a high risk of arbovirus transmission in Kinshasa [12,13]. Further, the survey showed that both species are well established in the capital of DRC.

In West Africa, *Ae. aegypti* is characterized as anthropophilic, with a bimodal diurnal biting behavior [14]. *Ae. albopictus* also voluntarily feeds on human blood in West Africa. Few studies in West Africa have investigated the indoor and outdoor biting and resting behaviors of both *Aedes* species [14]. Knowledge of the indoor and outdoor biting and resting behaviors is a prerequisite for implementing vector control measures. Therefore, this study aimed to assess the host-seeking and resting behaviors of female *Ae. aegypti* and *Ae. albopictus* and to determine their densities in four communes of Kinshasa, DRC.

## 2. Materials and Methods

***Study sites.*** The study was conducted in Kinshasa, the capital of the DRC, where we selected four different communes: Kalamu, Lingwala, Mont Ngafula, and Ndjili. The study localities were the same as those selected in the study by Wat’senga et al. [12], who assessed the infestation rates of immature *Aedes*. Briefly, the commune of Kalamu (Funa district) is a residential area located in the center of Kinshasa. Its main economic activity is technical service provision. It has an estimated population density of 47,000 persons/km^2^. Lingwala (Lukunga district) is a commune in the center of the city and has many informal markets. Its population density is estimated to be 33,000 persons/km^2^. Mont Ngafula is situated in the south of the city and is a semi-urban area with an estimated population density of 730 persons/km^2^. The economy relies mainly on agriculture and trade in agriculture products with Kinshasa city. Mont Ngafula is characterized by unplanned urbanization, with typically deficient water supply and wastewater disposal systems. Ndjili (Tshangu district) is a peri-urban commune in the east of the city with many informal economic activities, specifically vehicle repair shops. Most of the houses (97%) have a water supply system, but the quality and amount of available water is deficient. The population density of this area is estimated to be 39,000 persons/km^2^.

Kinshasa, with an estimated population of 12 million people, is characterized by a tropical climate, with a rainy season from October to May and a dry season from June to September. The average temperature varies between 18 °C and 32 °C.

***Sampling adult mosquitoes.*** Two entomological surveys were implemented: survey one was conducted in July 2019, coinciding with the dry season, and survey two was implemented in February 2020 during the rainy season. Mosquitoes were collected using three different adult collection methods targeting different aspects of adult mosquito behavior. BG-Sentinel 2 (Biogents AG, Regensburg, Germany), trap with the BG-lure^®^ (Biogents AG, Regensburg, Germany) was used to collect host-seeking adult female mosquitoes. The BG Lure^®^ is an artificial scent that mimics human odors and is designed to attract anthropophilic mosquitoes. As the BG-Sentinel 2 attracts host-seeking female mosquitoes, we used it as a proxy for the mosquito ’s biting activity; BG-GAT (Biogents AG, Regensburg, Germany) was used to collect gravid female mosquitoes seeking a larval habitat. BG-GAT is an artificial oviposition site that attracts gravid female mosquitoes. Mosquitoes enter a transparent chamber through the black funnel on top of the trap where they are caught. The prokopack aspirator collects resting mosquitoes. We selected five fixed houses in each study location. One BG-Sentinel 2 with BG-lure^®^ was put inside the house and one was put outside the house. The traps ran from 6 h to 21 h and catch bags were collected every three hours to identify the peaks of the host-seeking activities of the mosquitoes. Mosquitoes were collected over five nights. BG-GATs were used in five houses (different from those selected for BG-Sentinel 2 collections). One BG-GAT was put inside the house and another was put outside the house. Mosquitoes were collected on a daily basis, over the same period as the BG-Sentinel 2 collections. Resting mosquitoes were collected using the prokopack aspirator [15]. Each day, 10 houses were screened for 15 min by two collectors. One collector screened the sitting room, the bedroom, and the kitchen, while the other collector screened outdoors, including nearby vegetations and the exterior walls of the house. Collections were conducted over five days, with different houses being selected each day, resulting in 50 houses per commune per survey.

***Sample processing*.** Mosquitoes were morphologically identified using standard morphological identification keys [16,17]. *Aedes aegypti* and *Ae. albopictus* were identified up to the species level. Other mosquitoes were identified up to the genus level.

***Data analysis*.** Three different female adult *Aedes* indices were calculated based on the prokopack collections. Indices were calculated separately for each survey (i.e., the dry and the wet seasons), commune, and female *Ae. aegypti* and *Ae. albopictus*. Indoor and outdoor collections were combined.

Adult House Index (AHI): percentage of houses with at least one adult female *Aedes* mosquito.Female adult mosquito density (FAD): number of female *Aedes* mosquitoes collected divided by the number of houses with at least one adult female *Aedes* mosquito, i.e., adult female *Aedes* density in the houses where *Aedes* mosquitoes were found.Adult Breteau Index (ABI): number of adult female *Aedes* mosquitoes per 100 houses inspected.

To investigate the effect of the commune, season (dry = July 2019, wet = February 2020), time of collection (five three-hour-long timepoints during the day), and place of collection (indoor and outdoor) on the collection of female *Ae. aegypti* and *Ae. albopictus*, we developed a generalized linear model (GLM) with negative binomial distribution per species, with the number of female mosquitoes as the response variable and the commune, season, time of collection, and place of collection as explanatory variables (R package MASS, function glm.nb). The regression coefficients were exponentiated to provide the incidence rate ratio (IRR), where the incidence rate refers to mosquito density.

## 3. Results

Overall, we collected 1321 *Ae. aegypti*, 228 *Ae. albopictus*, 195 *Anopheles* spp., 21,972 *Culex* spp. and six *Mansonia* spp. across all collection methods and surveys (Table 1). Most mosquitoes were collected using the prokopack aspiration method (81%), followed by the BG-Sentinel 2 (17%) and BG-GAT (2%) collection methods.

### 3.1. The Effects of Commune, Season, Place of Collection, and Time of Collection on the Collection of Female Aedes aegypti and Aedes albopictus

Based on the BG-Sentinel 2 collections, most *Ae. aegypti* mosquitoes were collected in Mont Ngafula (Figure 1A and Table 2). We observed an 88% lower vector density in the dry season compared with the rainy season (IRR = 0.12, *p* < 0.0001). Likewise, there was an 87% lower vector density indoors compared with outdoors (IRR = 0.13, *p* < 0.0001) (Figure 1G; Table 2). *Aedes aegypti* showed a clear peak in host-seeking activity i.e., we observed a significant increase in the number of host-seeking mosquitoes from 6 h onwards, with a peak at 15–18 h (IRR = 10.43, *p* < 0.0001), after which the collections decreased (Figure 1J, Table 2). Significantly fewer *Ae. aegypti* were collected in Lingwala compared with Mont Ngafula using prokopack (Figure 1B, Table 2). As with the BG-Sentinel 2 collections, fewer *Ae. aegypti* mosquitoes were collected in the dry season (July 2019) compared with the rainy season (February 2020) (IRR = 0.04, *p* < 0.0001), and more mosquitoes were collected outdoors than indoors (Figure 1E,H, Table 2). No differences were detected in BG-GAT collections of *Ae. aegypti* between communes or between seasons. However, we observed 88% lower vector density indoors compared with outdoors (IRR = 0.12, *p* < 0.0001) using BG-GAT (Figure 1, Table 2).

No differences were observed in the numbers of *Ae. albopictus* collected using the BG-Sentinel 2 method in Mont Ngafula and Lingwala (IRR = 0.77, *p* = 0.4820), whereas we observed at least 77% lower mosquito densities in Kalamu and Ndjili compared with Mont Ngafula (Figure 2A, Table 3). The peak host-seeking activity of *Ae. albopictus* was less obvious than for *Ae. aegypti*, although we collected 3.4 times more *Ae. albopictus* between 15 h and 18 h compared with 6–9 h (IRR = 3.4, *p* = 0.0239). No differences were found between the communes based on the prokopack collections (Figure 2B, Table 3). Similar to *Ae. aegypti*, more *Ae. albopictus* were collected in the rainy season and outdoors using the prokopack collection method. Further, most *Ae. albopictus* were collected in Mont Ngafula using BG-GAT. We observed 88% lower *Ae. albopictus* density in the indoor versus outdoor BG-GAT collections (IRR = 0.22, *p* = 0.0001) (Figure 2, Table 3).

### 3.2. Adult Female Aedes aegypti and Aedes albopictus Indices

The adult female indices for *Ae. aegypti* were higher in the rainy season than in the dry season (Table 4). At least one adult female *Ae. aegypti* mosquito was collected in more than 55% of the houses in Mont Ngafula and Ndjili. However, this was only 27% in Lingwala. The density of *Ae. aegypti* in houses where the species was found ranged from 2.58 to 4.43 mosquitoes per positive house in the rainy season and from 1 to 1.2 mosquitoes per positive house in the dry season. The Adult Breteau Index was 190.77 mosquitoes per 100 houses in the rainy season and decreased to 6.03 mosquitoes in the dry season. Overall, the adult female indices for *Ae. albopictus* were smaller than for *Ae. aegypti* (Table 4). The indices for the four communes combined were smaller in the dry season than in the rainy season, except for the density of *Ae. albopictus* in positive houses (FAD = 1.75 and 1.53 mosquitoes per positive house in the dry and rainy season, respectively) (Table 4).

## 4. Discussion

We used different adult collection methods to obtain basic information on the behaviors and densities of *Ae. aegypti* and *Ae. albopictus* in Kinshasa, DRC. In our study, both *Aedes* species were exophagic, exophilic, and sought breeding sites outdoors. The house infestation rate, as measured using the AHI, exceeded 55% for *Ae. aegypti* in all communes except Lingwala, where at least one specimen of one of the species was caught in only 27% of the houses. Generally, *Ae. albopictus* had a lower AHI in this study. Overall, the adult densities of both species were low, with a surprisingly low number of *Ae. albopictus* collected using the different methods. This is in contrast to the chikungunya outbreak in Kongo Central (Matadi), during which the *Ae. albopictus* density was very high and the species was incriminated as the main vector driving the outbreak [9].

More host-seeking and resting adult female *Ae. aegypti* and *Ae. albopictus* were collected during the rainy season than the dry season, corroborating the results of the survey of the immature stages in Kinshasa [12]. The impact of the rainy season on adult mosquito density was not observed in the BG-GAT collections. This is likely due to the fact that during the rainy season, BG-GAT has to compete with the large number of larval breeding habitats present in the study localities, as shown by Wat’senga et al. [12]. In Kinshasa, temperature is not a limiting factor in the development of both *Aedes* species [18], although temperature and humidity play critical roles in the population dynamics of both species by impacting *Aedes* mortality and, hence, pathogen transmission [19]. The differences in densities between the dry and rainy seasons seem to be the result of the rainfall and the presence of water-containing larval habitats. Indeed, the most productive larval habitats during the dry season were outdoor water storage containers, whereas in the rainy season, rubbish and tires were key outdoor habitats [12]. Likewise, *Ae. aegypti* densities were positively linked to rainfall in West Africa [14]. However, this was less obvious for *Ae. albopictus*, which was found to be equally abundant during both the wet and dry seasons in Yaoundé [20].

The obvious exophagic and exophilic behavior of *Ae. aegypti* in Kinshasa is in clear contrast with the situation in Asia and Latin America, where *Ae. aegypti* mosquitoes typically feed and rest indoors. Studies conducted in Panama [21], Costa Rica [22], Dominican Republic [23], and Trinidad and Tobago [24] evidenced the exclusive indoor resting behavior of *Ae. aegypti*. A study carried out in Sri Lanka revealed that the resting behaviors of the two *Aedes* vector species varied: adult *Ae. aegypti* mosquitoes were highly endophilic, whereas *Ae. albopictus* demonstrated exophilic behavior, suggesting that a domestic environment with high human–vector contact, especially in urban areas, provides suitable breeding and resting sites for *Ae. aegypti* mosquitoes, while *Ae. albopictus* is found mainly among the vegetations in rural and suburban areas of this region [25]. Egid et al. [14] recently reviewed the biting and resting location of *Ae. aegypti* in Western Africa: the species was characterized as endophilic and endophagic in Côte d’Ivoire and Niger, but displayed a more marked exophagic behavior in Ghana [14]. In Kenya, *Ae. aegypti* was found to primarily bite and rest outdoors [26,27]. Studies assessing the behavior of *Ae. albopictus* in Africa are few. However, this species seems to be primarily exophagic and exophilic in Asia, Latin America, and Islas Reunión [28], which is in line with our observations.

In Kinshasa, *Ae. aegypti* showed unimodal host-seeking behavior within the observation period of 6–21 h, with an increase in the number of host-seeking female mosquitoes from 6 o’clock in the morning until 15–18 h in the afternoon, after which collections dropped. This is in contrast to observations in West Africa, where the species showed a bimodal biting rhythm during the day, with the first peak in the morning and another larger peak of biting activity around sunset. Likewise, in Sri Lanka, *Ae. aegypti* demonstrated a bimodal host-seeking cycle, with a minor peak between 5 h and 9 h and a major peak between 13 h and 19 h in the afternoon [25]. However, we could not reliably assess the day host-seeking rhythm for *Ae. albopictus* due to the small number of mosquitoes collected. In Cameroon, *Ae. albopictus* had one activity peak between 15 h and 19 h [29], whereas in Sri Lanka, the species’ biting activity was bimodal, with two equally dominant peaks in the morning between 5 and 11 h and in the afternoon between 14 h and 19 h [25].

Classically, two morphological subspecies are recognized within *Ae. aegypti: Aedes aegypti aegypti* and *Aedes aegypti formosus* [30]. *Aedes aegypti formosus* is considered the ancestral sylvatic form that primarily breeds in non-human-disturbed habitats and prefers non-human blood meals. However, there is evidence that this subspecies is increasingly present in human habitats [31]. *Aedes aegypti aegypti* is the ‘domestic’ form and prefers to bite humans, breeds in human-made habitats, and has spread throughout the tropics and subtropics. In the current study in Kinshasa, DRC, we observed *Ae. aegypti* as primarily biting and resting outdoors, a behavior typically linked to *Ae. aegypti formosus*. The actual distribution of both subspecies in Africa is uncertain: *Aedes aegypti formosus* is considered to be widespread in Africa, including in DRC, whereas the occurrence of *Ae. aegypti aegypti* in Africa seems to be more restricted [31,32]. Studies in Senegal showed that *Aedes aegypti aegypti* is prevalent in coastal cities, whereas *Aedes aegypti aegypti* and *Aedes aegypti formosus* are observed sympatrically along forest edges [33]. However, a clear correlation between mosquito morphology and ecology is not well established and further investigations are warranted in DRC [2].

This study has further demonstrated the efficacy of prokopack aspiration, which collected 81% of all adult *Ae. aegypti*, followed by BG-Sentinel 2 traps (17%) and BG-GAT (2%). In contrast, previous studies have found that BG-Sentinel traps are more effective than other tools [26,34]. The relatively large number of *Ae. aegypti* mosquitoes collected using prokopack aspiration suggests that this tool can significantly complement other *Ae. aegypti* and *Ae. albopictus* surveillance tools in Kinshasa. During the current study, *Ae. aegypti* was predominant in the study communes. Recently, one study carried out in Kinshasa demonstrated that both species were widely distributed, but not homogeneously, with greater *Ae. albopictus* prevalence in areas with more vegetations [35]. On the other hand, in the Republic of the Congo, both species bred together throughout the country, with *Ae. albopictus* present in all localities except Brazzaville and Pointe Noire [36]. In contrast, *Ae. aegypti* was predominant in Cuba, and the two species were not found breeding together in urban areas in Havana [37].

The control of *Aedes*-borne pathogens mainly involves controlling vectors at different life stages, as prevention through immunization is not yet programmatically available, except for the control of yellow fever. Larval source management is one of the key interventions, whereas adult *Aedes* control strategies are primarily considered for outbreak control. However, the epidemiological impact of adult vector control is uncertain (see e.g., [18]). In Kinshasa, adult vector control methods should target mosquitoes outdoors due to the marked exophagic and exophilic behavior of the two vector species *Ae. aegypti* and *Ae. albopictus*. However, both species are resistant to commonly used insecticides such as Permethrin, Lambda-cyhalothrin, and Malathion (EZM, personal communication). Indeed, insecticide resistance in both *Ae. aegypti* and *Ae. albopictus* is a growing problem in many African countries [38,39,40,41,42,43,44,45] and is expected to limit the use of insecticide-based *Aedes* control strategies. The use of larval source management should therefore be explored further. These results indicate that *Aedes* control strategies in Kinshasa need to target outdoor spaces to destroy or reduce larval habitats. The use of larvicide-based strategies is questionable because the principal breeding sites identified in Kinshasa are temporary and subject to continuous water changes [10]. Due to this complexity in control strategies, a longitudinal study describing the fine-scale seasonality of *Aedes* infestation could provide evidence in support of temporal differentiation of control measures.

## 5. Conclusions

This study identified *Ae. aegypti* and *Ae. albopictus* in Kinshasa as exophagic and exophilic, with a unimodal diurnal host-seeking rhythm in the former species. Hence, adult vector control strategies will need to target mosquitoes outdoors. Further, information on the correlation between adult vector density, rainfall, and the presence of the most productive breeding habitats per species is needed to understand how these factors shape the abundance and seasonality of these species.

## Figures and Tables

**Figure 1 tropicalmed-08-00207-f001:**
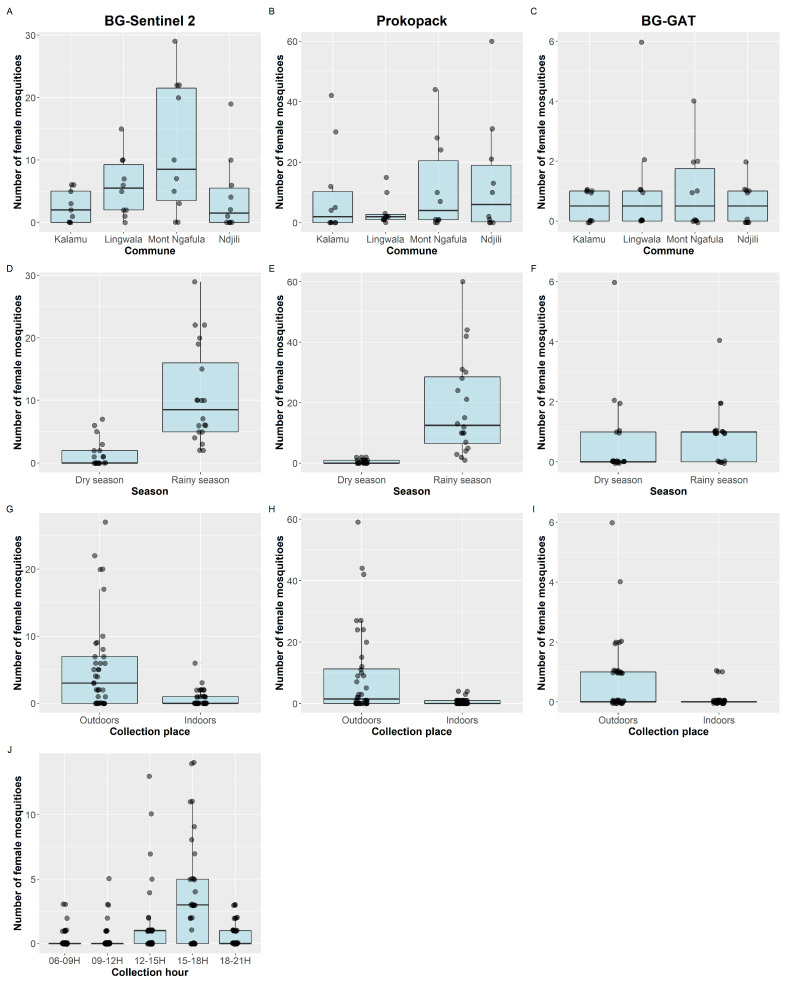
Number of female *Aedes aegypti* mosquitoes collected per day (dots), summarized by commune (**A**–**C**), season (collection surveys) (**D**–**F**), collection places (**G**–**I**), and collection time (**J**). The boxplots provide an overview of the medians and interquartile ranges. Column 1 provides data from the BG-Sentinel 2 collections, column 2 from the prokopack collections, and column 3 from the BG-GAT collections. Note. The scales of the *y*-axes are adjusted in each panel to enable interpretation of the data.

**Figure 2 tropicalmed-08-00207-f002:**
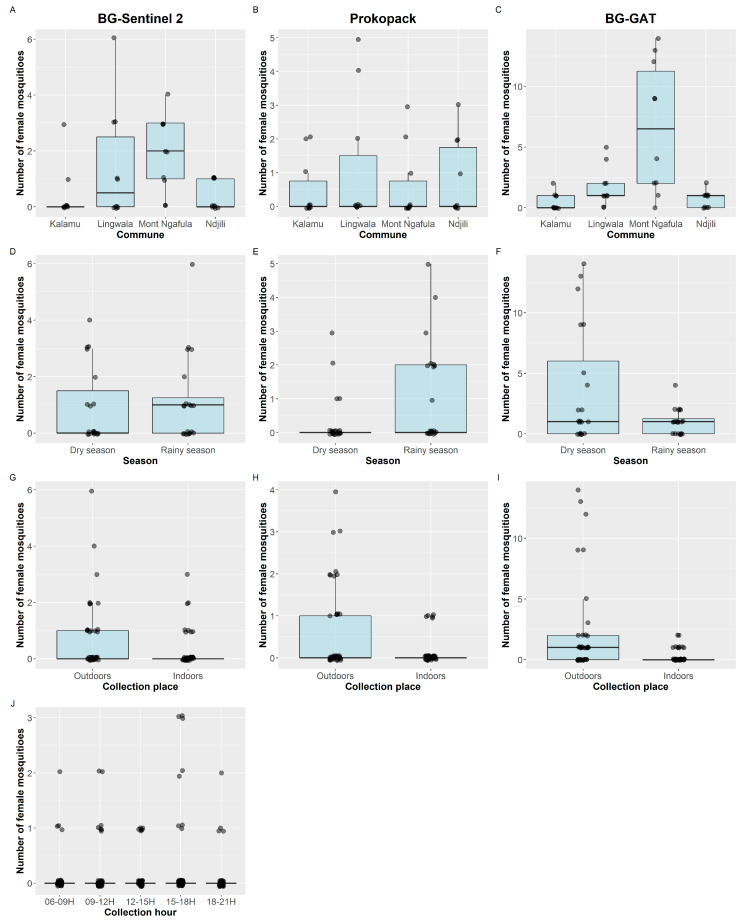
Number of female *Aedes albopictus* mosquitoes collected per day (dots), summarized by commune (**A**–**C**), season (collection surveys) (**D**–**F**), collection place (**G**–**I**), and collection time (**J**). The boxplots provide an overview of the medians and interquartile ranges. Column 1 provides data from the BG-Sentinel 2 collections, column 2 from the prokopack collections, and column 3 from the BG-GAT collections. Note. The scales of the y-axes are adjusted to the collections to enable interpretation of the data.

**Table 1 tropicalmed-08-00207-t001:** Overview of the number of collected mosquitoes per species or genus across all collection methods and surveys.

Trap	Sex	*Aedes * *aegypti*	*Aedes * *albopictus*	*Anopheles * spp.	*Culex * spp.	*Mansonia * spp.
BG-Sentinel 2	female	241	41	15	1354	0
	male	56	15	19	2344	2
Prokopack	female	384	30	109	9317	3
	male	604	31	51	8669	0
BG-GAT	female	32	95	1	137	1
	male	4	16	0	151	0
Total	female	657	166	125	10,808	4
	male	664	62	70	11,164	2

**Table 2 tropicalmed-08-00207-t002:** The effect of commune, season (survey), place of collection, and time of collection on the collection of female *Aedes aegypti* based on a GLM with negative binomial distribution. *p* Values lower than 0.05 are highlighted in bold.

Trap	Fixed Effects		Incidence Rate Ratio	*p* Value
BG-Sentinel 2	Intercept		0.2081	**<0.0001**
	Health Zone (ref = Mont Ngafula)	Kalamu	0.2074	**<0.0001**
		Lingwala	0.5307	**0.0050**
		Ndjili	0.3311	**<0.0001**
	Season (ref = wet)	dry	0.1206	**<0.0001**
	Trap location (ref = outdoors)	Indoors	0.1316	**<0.0001**
	Collection time (ref = 6–9 h)	9–12 h	1.1053	0.8117
		12–15 h	3.6872	**0.0003**
		15–18 h	10.4284	**<0.0001**
		18–21 h	2.0563	0.0581
Prokopack	Intercept		1.8938	**0.00542**
	Health Zone (ref = Mont Ngafula)	Kalamu	0.7598	0.37400
		Lingwala	0.4456	**0.01731**
		Ndjili	1.2236	0.49672
	Season (ref = wet)	Dry	0.0369	**<0.0001**
	Trap location (ref = outdoors)	Indoors	0.0723	**<0.0001**
BG-GAT	Intercept		0.1396	**<0.0001**
	Health Zone (ref = Mont Ngafula)	Kalamu	0.5458	0.36954
		Lingwala	1.3274	0.60939
		Ndjili	0.8935	0.85716
	Season (ref = wet)	Dry	0.4696	0.08760
	Trap location (ref = outdoors)	Indoors	0.1227	**0.00115**

**Table 3 tropicalmed-08-00207-t003:** The effects of commune, season (survey), time of collection, and place of collection on the collection of female *Aedes albopictus* based on a GLM with negative binomial distribution. *p* Values lower than 0.05 are highlighted in bold.

Trap	Fixed Effects		Incidence Rate Ratio	*p* Value
BG-Sentinel 2	Intercept		0.0365	**<0.0001**
	Health Zone (ref = Mont Ngafula)	Kalamu	0.2354	**0.0106**
		Lingwala	0.7674	0.4820
		Ndjili	0.2111	**0.0063**
	Season (ref = wet)	Dry	0.7219	0.3326
	Trap location (ref = outdoors)	Indoors	0.4073	**0.0125**
	Collection time (ref = 6–9 h)	9–12 h	1.8959	0.2744
		12–15 h	1.2287	0.7452
		15–18 h	3.4033	**0.0239**
		18–21 h	1.0060	0.9928
Prokopack	Intercept		0.0770	**<0.0001**
	Health Zone (ref = Mont Ngafula)	Kalamu	0.9743	0.97119
		Lingwala	1.7898	0.38635
		Ndjili	1.4317	0.59804
	Season (ref = wet)	Dry	0.2930	**0.01720**
	Trap location (ref = outdoors)	Indoors	0.1911	**0.00332**
BG-GAT	Intercept		0.3931	**0.005862**
	Health Zone (ref = Mont Ngafula)	Kalamu	0.1053	**<0.0001**
		Lingwala	0.3079	**0.002825**
		Ndjili	0.1678	**0.000384**
	Season (ref = wet)	Dry	1.7298	0.110098
	Trap location (ref = outdoors)	Indoors	0.2224	**0.000109**

**Table 4 tropicalmed-08-00207-t004:** Female *Aedes* mosquito indices per species and season (combined indoor and outdoor collections) in the four study sites in Kinshasa. (Dry season = July 2019 and rainy season = February 2020). Indices were based on the prokopack collections (50 h/commune/survey).

Species	Index	Season	Kalamu	Lingwala	Mont Ngafula	Ndjili	Total
*Aedes * *aegypti*	AHI	Dry	0	10.00	6.00	6.12	5.53
		Rainy	42.00	26.67	58.00	62.00	47.69
	FAD	Dry	0	1.20	1.00	1.00	1.09
		Rainy	4.43	2.58	3.90	4.35	4.00
	ABI	Dry	0	12.00	6.00	6.12	6.03
		Rainy	186.00	68.89	226.00	270.00	190.77
*Aedes * *albopictus*	AHI	Dry	4.00	0	2.00	2.04	2.01
		Rainy	2.00	13.33	10.00	6.00	7.69
	FAD	Dry	1.50	N/A	1.00	3.00	1.75
		Rainy	2.00	1.83	1.00	1.67	1.53
	ABI	Dry	6.00	0	2.00	6.12	3.52
		Rainy	4.00	24.44	10.00	10.00	11.79

## Data Availability

The data presented in this study are available on request from the corresponding author.

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
