# Peer review of "Behavior of Adult Aedes aegypti and Aedes albopictus in Kinshasa, DRC, and the Implications for Control"

_tropicalmed, 2023, doi:10.3390/tropicalmed8040207_

Round 1

Reviewer 1 Report

General comments-

Understanding the ecology and behavior of vector mosquitoes is vital for successful control interventions. Aedes aegypti and Ae. albopictus are two important vectors of dengue and other arboviruses, but there is a lack of local information on these species in the Democratic Republic of Congo (DRC). This study presents data and observations from field surveys of mosquitoes in Kinshasa, DRC using three trap types which focus on catching host-seeking, gravid and resting mosquitoes. The authors compare patterns of abundance between the different trap types, different communes and indoor and outdoor locations, as well as comparing seasonal and diurnal patterns of abundance. They find that Ae. aegypti and Ae. are found at a much higher abundance in outdoor locations compared to indoors. They also show clear seasonal and locational patterns which could help to inform the timing and intensity of mosquito control programs. Overall, the paper is a useful contribution to the field with only a few minor changes required. My only main concern is that the authors are basing their claims about host-seeking/biting behavior on BG-trap data which is not equivalent to a live human, especially since traps did not appear to include CO2, an important host-seeking mosquito attractant.

Specific comments-

Line 31- rephrase- “biting rhythm” is misleading since biting was not measured directly. See further comments below. Also rephrase “unimodal”- since mosquitoes were not collected across a 24 hr period, you cannot rule out a bimodal pattern

Line 87- provide more information on the BG-sentinel traps- e.g. what trap model was used? It could also help to further describe the BG lure (a lactic acid formulation which mimics human odors) and elaborate on its purpose.

Line 88 – a brief description of the GAT would be useful too

Line 105- Please clarify if male mosquitoes were included or excluded from the analyses

Line 138 – “biting activity” is misleading since you did not measure biting directly, only a proxy for biting activity by collecting mosquitoes attracted to BG traps with human odor mimics.

Line 147 – Rephrase- change “a 88% reduction in vector density” to “an 88% lower vector density”

Line 152 – change to “observed at least a reduction of 77% of the mosquito density” to “observed a 77% lower mosquito density”. Also rephrase other sections below where “reduction” is used.

Figures 1 and 2 – increase text size for all panels. I am also confused why many data points appear below zero.  This is especially obvious for figure 2J, where about half the data points are below zero on the y axis- how is this possible when the data show mosquito counts?

Figure 1 and 2 legend- specify if the data includes males or just females

Line 245- Approximately what time is sunrise? Is it possible that there is a peak before 6 that the authors have missed due to not monitoring for a full 24 hr? If so, this paragraph should be rephrased to acknowledge the possibility that a peak could have occurred before monitoring took place.

Author Response

Please see the attachmant

Reviewer 2 Report

It is a relevant work that brings a lot of information that can be of great value for the monitoring and control of mosquito vectors. There are minor revisions needed throughout the manuscript.
